# Smart thermosensitive poloxamer hydrogels loaded with Nr-CWs for the treatment of diabetic wounds

Jian Wang[1☉], Bingkun Zhao[1☉], Lili Sun[1], Liqun Jiang[2], Qiang Li[1]*, Peisheng Jin[1]*

1 Department of Plastic Surgery, Affiliated Hospital of Xuzhou Medical University, Xuzhou, Jiangsu, China,
2 Xuzhou Medical University, Xuzhou, Jiangsu, China

☉ These authors contributed equally to this work.
* 83409322@qq.com (QL); 100000401006@xzhmu.edu.cn (PJ)

**Data Availability Statement:** All relevant data are within the paper and its Supporting information files.

**Funding:** This work was supported by the [National Natural Science Foundation of China] under Grant

## Abstract

The treatment of diabetic wound is a focus issue. At present, the Nocardia rubra cell wall skeleton (Nr-CWS) has been proved proven to promote angiogenesis and wound repair. Unfortunately, the high-glucose diabetic wound environment makes many drugs unable to be released effectively, and soon be removed. Smart thermosensitive poloxamer hydrogel (TH) is an ideal and adjustable drug delivery platform compatible with most living tissues. Here, a multifunctional composite thermosensitive hydrogel was developed. A mixture of poloxamers 407 and 188 as the gel matrix, and then it was physically mixed with Nr-CWS. The delivery vehicle not only controlled its release stably, preventing degradation in vitro, but also showed good affinity in vitro. In vivo, compared with thermosensitive poloxamer hydrogel alone or the direct use of Nr-CWS, the thermosensitive poloxamer hydrogel loaded with Nr-CWS promoted the proliferation of vascular endothelial cells effectively, resulting in increased expression of derma-related structural proteins and enhanced angiogenesis and wound healing. This study indicated that the angiogenesis and skin regeneration brought by Nr-CWS hydrogel are related to the activation of phosphatidylinositol 3 kinase and protein kinase B, Janus kinase/signal transducer and activator of transcription, and mitogen-activated protein kinase kinase/extracellular signal-regulated kinase signaling pathways.

## 1 Introduction

The occurrence of chronic diabetic wounds depends on many factors. Diabetic wounds are one of the most problematic complications of diabetes [1]. Their complex and bad condition leads to slow wound healing and even the risk of amputation. Despite the application of various drugs in the treatment of diabetes and a range of conventional therapies such as silver ion dressings and nanomaterials, for chronic wounds, recovery remains unsatisfactory, with approximately 28% of patients subjected to amputation and a mortality rate of 50–59% after 5 years [2]. Therefore, appropriate therapeutic drugs and treatment modalities are urgently needed.

[number 82172224], the [National Natural Science Foundation of Jiangsu Province] under Grant [number BK20201155] and the [Postgraduate Research & Practice Innovation Program of Jiangsu Province] under Grant [number KYCX21_2692].

**Competing interests:** Conflict of interest statement The authors declare that they have no competing interests.

Diabetic wound is a chronic inflammatory wound that is difficult to heal and has insufficient blood supply in some areas. Therefore, wound temperature is often higher than the normal body temperature [3]. Due to their high water content, different properties, and similarity to the native extracellular matrix, hydrogels are used as substrates for cell cultures, templates for tissue engineering, and carriers for drug and protein delivery. Due to the increase of inflammation temperature in the wound, poloxamer gel was stimulated by temperature and quickly changed to gel state. Based on its sol-gel transformation characteristics, poloxamer could be used as a drug carrier in the fields of burn treatment and transdermal drug delivery [4–6]. We studied different kinds of poloxamer and found that the single use of poloxamer could not carry out liquid gel mutual conversion at the required temperature, so we innovatively completed a task by using different proportions of two hydrogels. Hydrogel could absorb the exudates from the wound and improve the wound microenvironment due to it swelling ability [7]. Moist wound environment is conducive to cell proliferation and it promotes wound healing [8]. On this basis, as an immune enhancer, Nocardia rubra cell wall skeleton (Nr-CWS) has an extremely sensitive response to wound inflammation [9, 10].

Nr-CWS is a special cellular component of Gram-positive Nocardia rubra, which is partially composed of Nocardia polyphenolic acid, arabinogalactan, and mucin [11]. According to the latest report, Nr-CWS has been proven to accelerate healing of skin wounds by enhancing macrophage activation and angiogenesis. It also has potential antitumor properties, as confirmed in animal and clinical studies [12–14]. Although the full mechanism of its antitumor effect may be related to interferon [11], Nr-CWS has been proven to activate macrophages and increase local immune status [15]. Current research showed that the two key factors of wound healing in diabetes are 1) how to improve the local immune state and enhance macrophage activation [16] and 2) how to promote the formation of blood vessels and skin [17, 18]. In view of the superiority of Nr-CWS in macrophage activation and angiogenesis [19], it was chosen to treat diabetic wounds in the present work. However, due to the complexity of diabetic wounds, the direct use of drugs alone could soon degrade. Thus, the thermosensitive material hydrogel was used to load with Nr-CWS to promote diabetic wound healing.

This study aimed to investigate the effect of hydrogels loaded with Nr-CWS and the thermo-sensitive material of poloxamer on the healing of diabetic wounds. For the first time in full-layer skin repair, we tried to use P188 and P407 in combination to achieve a temperature controllable and wound protection effect. This method is simple and easy to use, and can be quickly applied to the clinic. We believe that in the healing process of diabetic wound, the use of combined immune drugs will be the first application on diabetic wound, and will promote the treatment of diabetic wound into a new stage. For the first time, Nr-CWS was loaded onto the thermosensitive material hydrogel to form Nr-CWS hydrogel (collectively called Nr-CWS TH). The morphology, rheology and cumulative release rate of the drug and hydrogel were detected and characterized by response surface methodology, scanning electron microscope, rheometer, particle size analyzer and animal thermal imager. Then, the wound healing, cell regeneration and mechanism of diabetes were evaluated by morphology, pathological sections, enzyme-linked adsorption and protein electrophoresis.

## 2 Materials and methods

### 2.1 Preparation of Nr-CWS TH

The gel response surface curve was prepared to obtain the best matching scheme. Appropriate poloxamer 188(P188) powder and poloxamer 407(P407) powder were added to distilled water and placed in 4˚C refrigerator for 24 h to completely dissolve. A transparent hydrogel loaded

with Nr-CWS was prepared by adding Nr-CWS powder to the thermosensitive hydrogel at 4°C. On the basis of clinical experience, the concentration of 60 μg/ml was used for this ratio.

## 2.2 Characterization of Nr-CWS TH

Rheological measurements of TH and Nr-CWS TH were conducted using a rheometers (Mars40, Thermofisher, USA) at different temperatures ranging from 5°C to 40°C. Elastic modulus and viscous shear modulus were also measured, and rheological curves were plotted with temperature. The micromorphology of dried samples was obtained by scanning electron microscopy (SEM, FEI, USA). Freeze-dried dehydrated samples were transected and over-gilded, followed by scanning observation. The size distribution of Nr-CWS was detected by laser particle scanning analyzer (Particle Metrix, Germany). A certain amount of P188/P407 composite freeze-drying gel was obtained and recorded as W dry. The hydrogel was soaked in a phosphate buffer solution at 37°C. After a period of time, the gel was dried, weighed, and recorded as W wet. All experiments were performed three times, and the average value was taken. Swelling ratio (SW) was calculated as follows: SW = (W wet–W dry) / W dry × 100%

## 2.3 Release profile of Nr-CWS from Nr-CWS TH hydrogels in vitro

At scheduled time points (0, 0.5, 1, 2, 4, 8, and 12h), the supernatants from Nr-CWS TH were collected and replaced with an equal volume of new culture medium. Then, Nr-CWS release was analyzed by measuring the absorbance of the supernatant.

## 2.4 Flow cytometry

HUVEC and fibroblast cells were cultured in TH and medium containing with or without Nr-CWS for 12 h. The treated cells were digested, centrifuged, and washed three times with PBS. Then, $1 \times 10^5$ HUVEC and cells treated by Nr-CWS, TH, and Nr-CWS TH were placed into tubes and added with 200μl binding buffer. The HUVEC and fibroblast cells were stained with 5μl Annexin V-FITC/PI (Keygen Biotech, China) and mixed well in the dark. The cells were detected using flow cytometry (Facs Canto II, BD, USA) after 15 min. Cell apoptosis was analyzed using FlowJo version 10 software.

## 2.5 Establishment of diabetic wound model

A diabetic wound model was established as described previously. Six week-old male Balb/c mice were purchased from Metrex, Xuzhou, China, and maintained under certain aseptic conditions. All animal studies were approved by the Animal Care and Ethics Committee of Xuzhou Medical University (project number: 202108w011). The diabetic mice were induced by intraperitoneal injection of 150 mg/kg streptozotocin (2% STZ, Sigma, Shanghai, China) after restriction of 12 h diet. The mice with blood glucose exceeding 300 mg/dl, with obvious characteristics of polyp, polyuria, and polyuria, were identified as diabetes. After the mice were anesthetized, a full-thickness skin defect with a diameter of 1.6 cm was formed on their back. All the 60 mice purchased were used as diabetic mouse model in accordance with the above method. They were divided into four groups on average: control group, blank gel group, drug group, and gel drug group. In the control group, PBS was used to wash the wound twice a day. In the blank gel group, empty thermosensitive material was used to smear the wound twice a day. In the drug group, PBS solution with a drug concentration of 60 μg/ml was used to clean the wound twice a day. In the gel drug group, gel with a drug concentration of 60 μg/ml was used to smear the wound. The treatments lasted for 14 days. The wound area was calculated by

Image J soft-ware (National Institutes of Health, USA). Wound healing index (%) was calculated using the following formula: = (1– unhealed wound area/original wound area) $\times$ 100%.

## 2.6 Release profile of Nr-CWS from Nr-CWS TH hydrogels in vivo

Cy5 (MedChem Express, China) was linked to nucleic acids or proteins through its reactive groups to label drugs. First, 60 µg of the drug was dissolved in 1 ml solvent to form a solution, and then the solution was incubated with Cy5 stain (1 µmol/L) at 37°C for 0.5h, shaken, and mixed at 4°C for 15 minutes. The mixed solution was passed through an 8000 molecular weight dialysis bag (MD10, solarbio, china), and the macromolecules and drugs left in the bag were used for subsequent animal experiments. Finally, it was injected into the wound surface. Nr-CWS was pre-labeled with the fluorescent dye Cy5 before drug injection. Bioluminescence imaging of mice was performed on the Xenogen ivis200 imaging system (Xenogen Corp.) to detect the efficiency of Cy5 staining. Finally, fluorescence intensity was detected at 649 nm.

## 2.7 Histological analysis

The new tissues around the wounds on days 6 and 14 were collected and fixed with 10% formalin. The formation of epidermis was observed by HE staining. Masson staining was used to determine the collagen content by Sirius red staining. The tissues were sectioned and then Masson stained using a three-step method. The first step was to use Regaud hematoxylin dyeing solution (1:2000, CST, USA) for 5–10 min, and the second step was to use Masson Ritchon red acid complex red solution (1:2000, CST, USA) for 5–10 min. Finally, the tissues were stained directly with aniline blue or light green liquid (1:5000, Abcam, UK) for 5 min. After elution, the images were observed and photographed under the microscope. Skin biopsies, including the wound and surrounding tissues, were harvested. Vessels were stained with a specific antibody against CD31 (1:50, CST, USA) and labeled red. Tumor necrosis factor $\alpha$ (TNF-$\alpha$) was marked in green. The nuclei were stained blue with DAPI dye. All images were captured by Camedia Master digital camera. (Olympus, USA).

## 2.8 ELISA

ELISA was performed to determine the production of IL-10 and IL-6. Skin biopsies from days 6 and 14 were performed in accordance with the protocol of each ELISA kit (Abcam, UK). The OD value of each well was recorded with the automatic microplate reader AMR-100 (Hangzhou Aosheng Instrument Co., Ltd., China).

## 2.9 Western blot analysis

The regenerated epidermis extracts were separated on SDS-polyacrylamide gels, and then transferred onto NC membranes (Pall, USA). The following primary antibodies were used: VEGF (1:500, proteintech, China), EGF (1:500, proteintech, China), GAPDH (1:2000, CST, USA), p-AKT (1:5000, Abcam, UK), AKT (1:5000, Abcam, UK), p-ERK (1:4000, proteintech, China), ERK (1:3000, proteintech, China), p-STAT3 (1:2000, Santa, USA), STAT3 (1:2000, Santa, USA), and GAPDH (1:2000, CST, USA).

## 2.10 Statistical analysis

Data were presented as mean ± standard deviation (SD). T-test was used for comparison between two groups, and SPSS 20.0 software (SPSS Inc., Chicago, Illinois, USA) was used for multiple comparisons between groups by using one-way ANOVA. The response surface curve

was designed by Design-Expert 10 (State-East, USA), and the influence of formula composition was analyzed by multiple regression. P < 0.05 indicated statistically significant difference.

## 3 Result

### 3.1 TH loaded with Nr-CWS maintains similar thermal-sensitive characteristics

The P407 and P188 purchased from MERYER in Shanghai are high-purity polymers with molecular weights (MWs) of 9840 and 12300, respectively. The gel response surface curve (Fig 1A) was drawn to obtain the best proportioning scheme to achieve appropriate gelation temperature for wound healing. The following linear equation was obtained from the multiple regression analysis of the influence of formula composition on the gelation temperature of blank hydrogel:

$GT = +119.190 – 6.259 \times {}^*P407 + 1.383^* \times P188 + 0.0689^* \times P407^* \times P188 + 0.0622^* \times P4072 – 0.149^* \times P188$. At 37˚C, 18% P407 and 4% P188 were chosen to form the gel. The particle size of Nr-CWS was measured to be about 520 nm before loading the drug (Fig 1B). As shown in Fig 1C, TH was a liquid state at 4˚C. However, it rapidly converted to hydrogel state after heating to 37˚C, and it melted to solution state again when the temperature returned to 4˚C. The results of Nr-CWS TH and TH are the same (Fig 1E). The temperature of 37˚C was selected because the body surface temperature is 37˚C. Next, the phase transition temperature of TH was further explored through rheological experiments. Storage modulus (G′) and loss modulus (G″) corresponded to changes in viscosity and elasticity, respectively. The G′ and G″ "values of TH increased quickly under the temperature of 25˚C–30˚C (Fig 1D), manifesting a solution-hydrogel phase transition. Therefore, it could easily form the hydrogel at 37˚C in wound. A similar process of solution-hydrogel-solution (sol-gel-sol) phase transitions could be observed under the same temperature after mixing Nr-CWS with TH (Fig 1F). After the hydrogels were loaded with Nr-CWS, the temperature of sol-gel compatibility of the hydrogel decreased to 20˚C. Thus, the gelation forming time was shortened (Fig 1F), possibly because the entanglement of P407 and P188 was disturbed after blending with Nr-CWS. The highest swelling rates of the TH group and the Nr-CWS TH group were 3.62 and 3.86, respectively (Fig 1G).

### 3.2 The safe porous structure persistently maintains the stability of the hydrogel

The results demonstrated highly arranged regular pore structure in the blank hydrogel (Fig 2A). The microscopic morphology of dehydrated TH-based hydrogel was studied by SEM. Hydrogel was used to carry the drug, as indicated by the red arrow in Fig 2B. HUVEC and fibroblast cells were cultured in TH and medium containing with or without Nr-CWS for 12 h to determine the safety of Nr-CWS hydrogel. The result of Annexin V+/PI- illustrated no obvious discrepancy among the four groups on early apoptotic rate via Facs Canto II. The results of the HUVEC and fibroblast cells cultured in normal medium were the same (Fig 2C–2F), implying that the essential component of TH in Nr-CWS hydrogel had no effect on HUVEC and fibroblast cell survival.

### 3.3 The slow release prolongs the duration of the drug's effect on the wound in vitro and in vivo

First, the release amount of Nr-CWS was tested in vivo. Cy5 stain was used to label the drugs, and a small animal live imager was utilized to measure the distribution and release of the gel

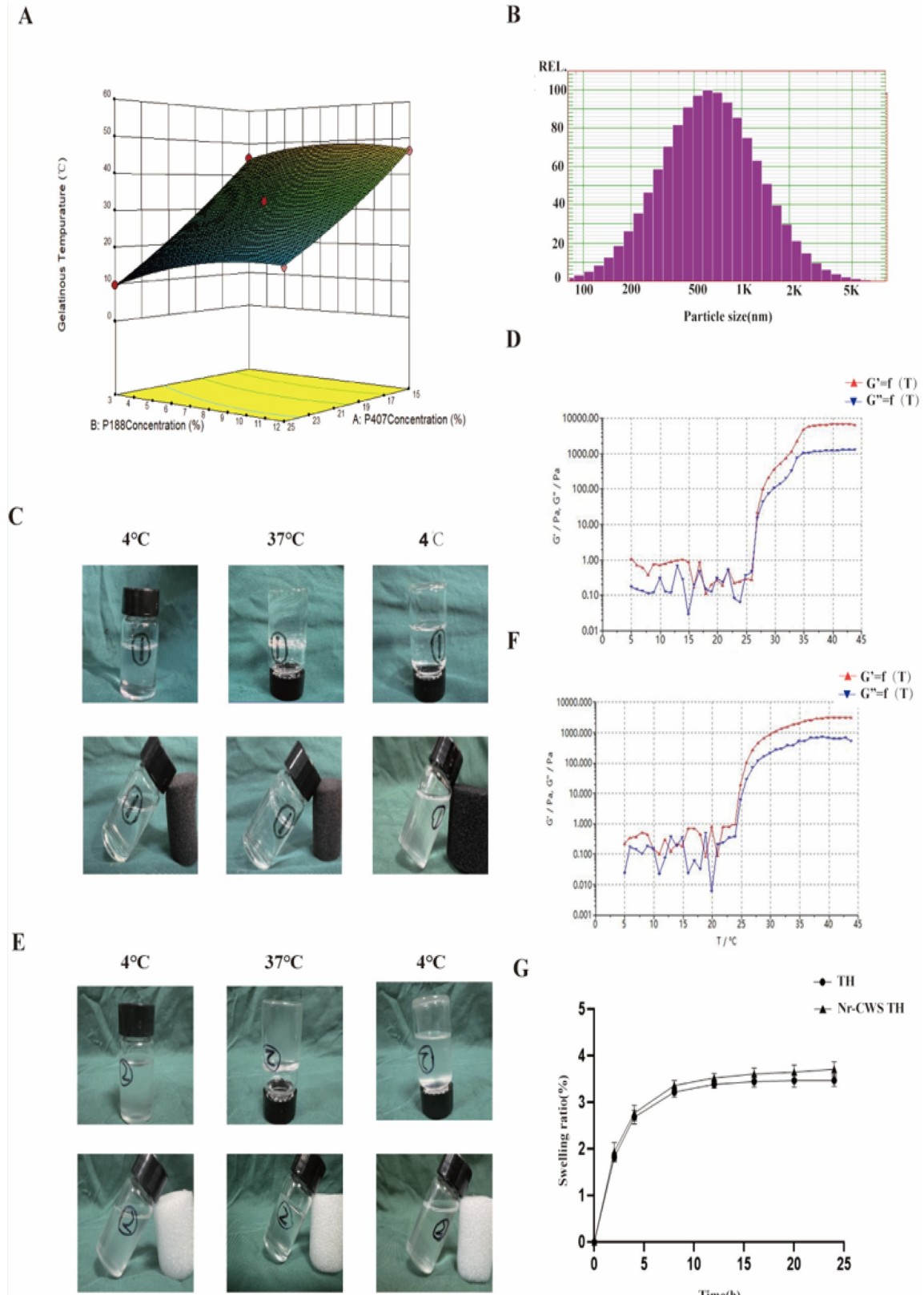

**Fig 1. The characterization of the hydrogel.** The characterization of the hydrogel. A) Impact of the formulation compositions on the gelation temperature (GT) of blank hydrogel. The higher concentration of P407, the lower GT of hydrogel. The GT of blank hydrogel

increased to a maximum and then decreased as a function of the Pol-2 concentration. B) The diameter and particle concentration of N-CWS were examined by NTA. C, E) Visualization of the state of TH and N-CWS TH respectively at different temperatures (4°C, 37°C and 4°C after 37°C). D, F) Storage (G′) and loss (G″) moduli as rheological markers in temperatures ranging from 5 to 45°C for TH and N-CWS TH hydrogels G) Swelling properties of N-CWS composite hydrogel.

drug in the Nr-CWS group and the Nr-CWS TH group within 12 h (Fig 3A and 3B). Then, the release amount of Nr-CWS TH was examined in vitro. Nr-CWS TH, which was in hydrogel state, was placed in the supernatant, and the absorbance of the supernatant at 0, 0.5, 1, 2, 4, 8, and 12 h was measured to record the release of Nr-CWS TH (Fig 3C). The release of hydrogel was found to be slow release, and it prolonged the effect of the drug on the wound.

### 3.4 The controlled release of Nr-CWS from Nr-CWS TH continuously promotes wound healing

The wounds of diabetic mice in different groups were observed on days 0, 3, 6, 9, 12, and 14 (Fig 4A). The Nr-CWS TH group had the highest healing rate on days 6 and 14 (Fig 4B and 4C). The wound healing rate increased to 96.8% within 14 days (Fig 4C) in this group. Similarly, the wound healing rates in the Nr-CWS and TH groups were lower than those in the TH group at 67.2% and 84.5%, respectively. Meanwhile, the blank gel group had slightly higher wound healing rate of 67.2% than the control group (60.2%).

### 3.5 Nr-CWS TH can accelerate epidermal regeneration and repair fibroblasts in wounds

The histological structures of the regenerated dermis were analyzed on days 6 and 14 (Fig 4D). As shown in Fig 4D, on day 6, only a little epidermis grew out in the control group and TH group, and it was hardly seen in the dermis. The arrangement of collagen fibers was loose and disordered. Histology revealed that compared with the control and TH groups, few inflammatory cells and neutrophils could be found in the wound treated with Nr-CWS TH during the 6-day treatment. The wound in the Nr-CWS TH group was thicker than that in the Nr-CWS group on day 6 days after treatment. The epidermis of the new granulation tissue was integrated and thick in the Nr-CWS and Nr-CWS TH groups. However, the Nr-CWS TH group showed fibroblast proliferation with adequate collagen deposition arranged in an orderly manner under the epidermis, as seen in Nr-CWS. All these findings indicated that significantly acute local inflammation was further reduced by Nr-CWS TH, thus accelerating wound healing. Moreover, Nr-CWS TH significantly promoted complete regeneration of epidermis and dermis. After 14 days, the wound of diabetic mice treated with Nr-CWS TH was covered the regenerated skin. The Masson-stained sections days 6 and 14 (Fig 4F) showed that the collagen fibers in the Nr-CWS TH group were arranged most neatly and the color was the deepest, indicating that the effect of Nr-CWS TH was the best in the repair of collagen fibers, followed by that of Nr-CWS.

### 3.6 Nr-CWS TH can reduce the inflammatory reaction of wound and accelerate wound healing

As previously mentioned, wound healing is a continuous and complex process. Thus, immunohistochemical evaluation of wound tissues was performed on days 6 and 14. TNF-α and IL-6 are typical pro-inflammatory factors. The growth of inflammation is fast at the beginning. On days 6 (Fig 5G) and 14 (Fig 5H), the IL-6 level in the Nr-CWS TH and Nr-CWS groups was much lower than that in the control and blank gel groups. The IL-6 level in the Nr-CWS

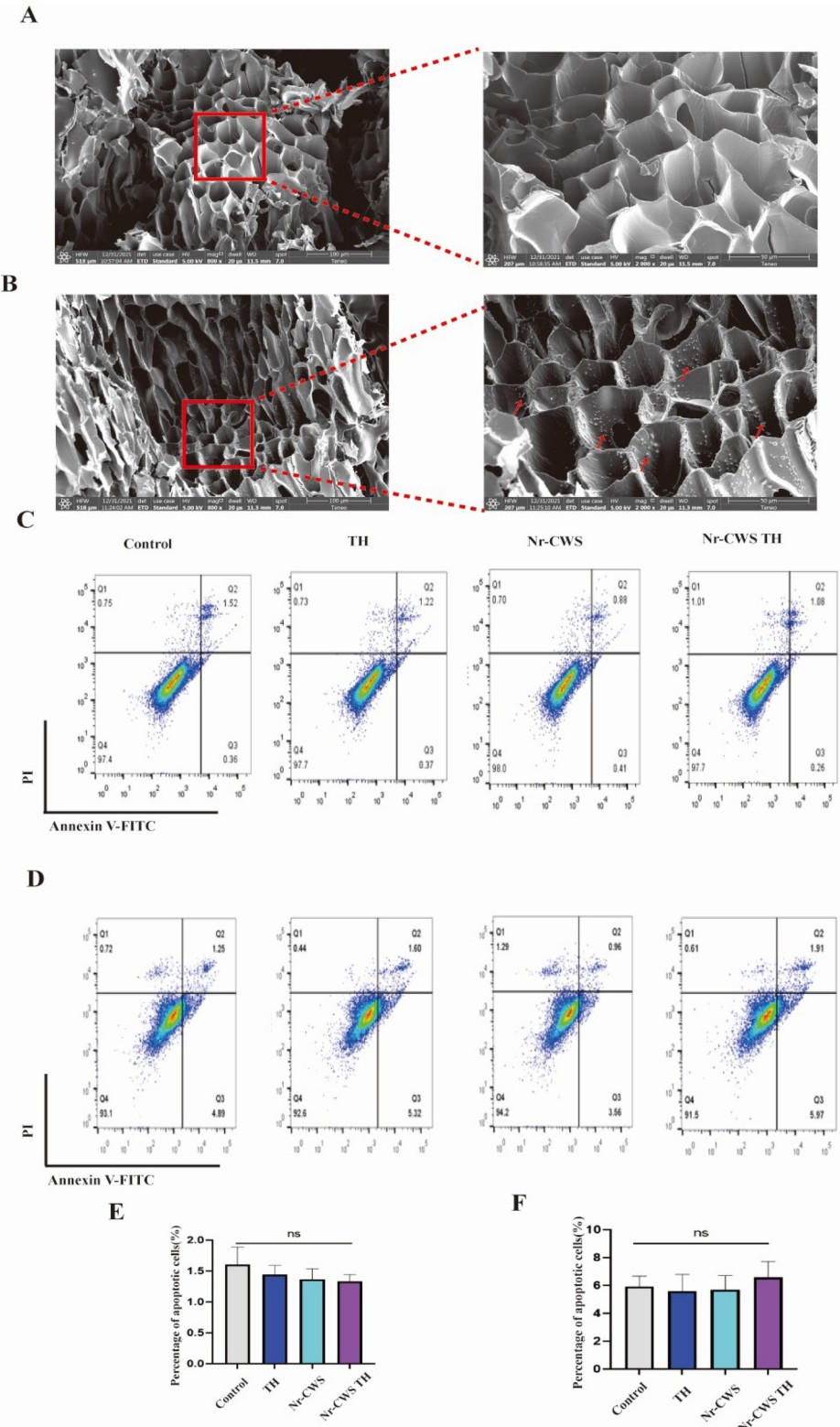

**Fig 2. The continuous release of N-CWS from N-CWS TH hydrogel in vivo and vitro.** Three-dimensional structure and safety of hydrogel A,B) The microstructure and safety of TH-hydrogel with/without N-CWS. The red arrow indicates the loaded N-CWS. Scale bar = 100 μm and Scale bar = 50 μm; C-F) The survival rate of HUVEC cells (C,E) and fibroblast cells (D,F) treated with/without N-CWS and TH using PI/annexin V-FITC staining and statistical results of early apoptosis rate. (n. s, nonsignificant).

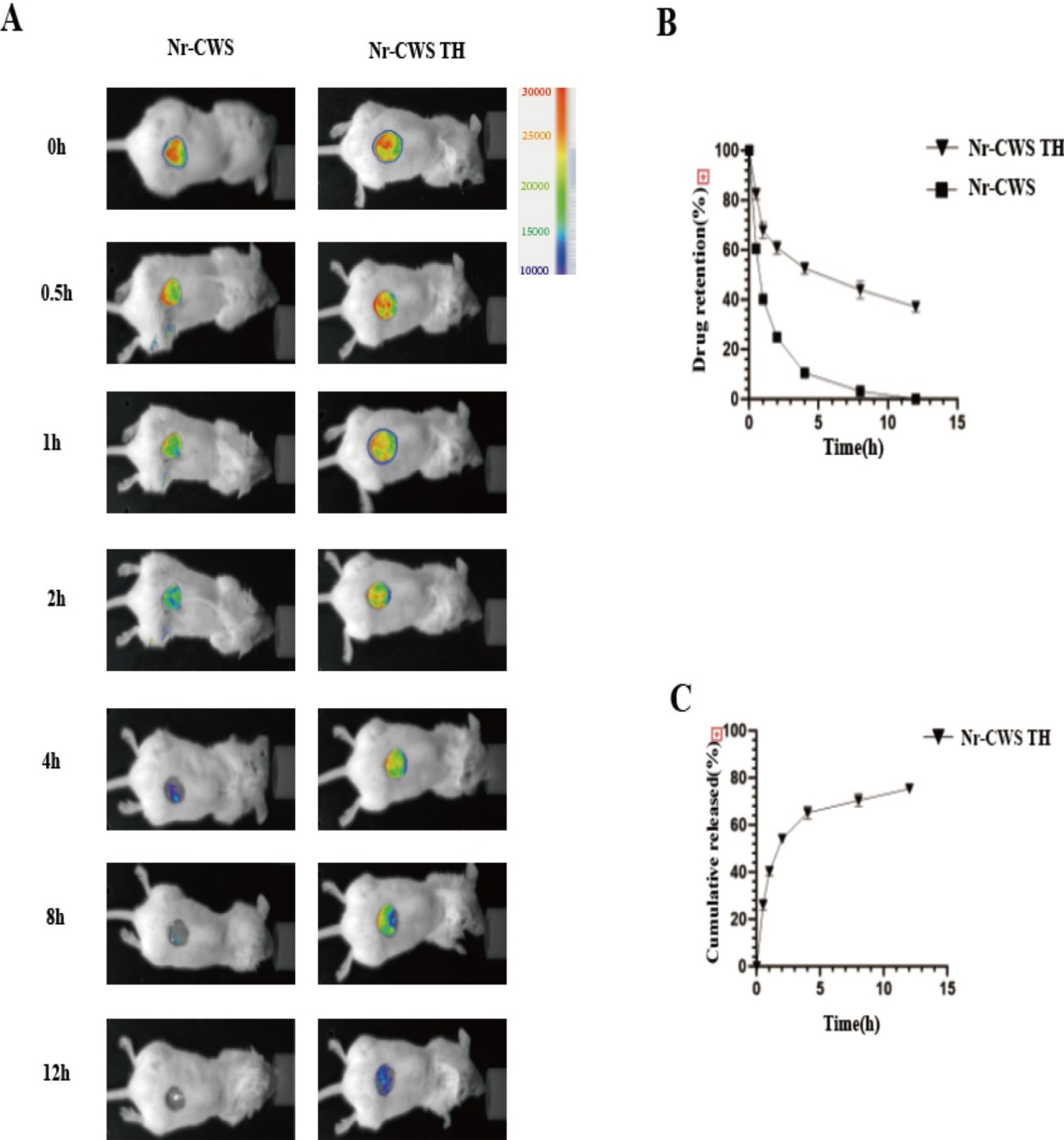

**Fig 3. The rate and condition of wound healing.** The continuous release of N-CWS from N-CWS TH hydrogel in vivo and vitro. A) After the wounds were formed, N-CWS stained with Cy5 were injected into mice. The continuous release of N-CWS from PBS and N-CWS TH hydrogel around the wound was analyzed by in vivo imaging. B) Statistical analysis of the continuous release of N-CWS in vivo; C) Release profile of N-CWS TH at different time points in vitro.

TH group was slightly lower than that in the Nr-CWS group, indicating that the drug-loaded gel played a very important role in drug release. IL-10 was also measured (Fig 5I and 5J), a factor that inhibits inflammation. The results found that the IL-10 level in the Nr-CWS TH group

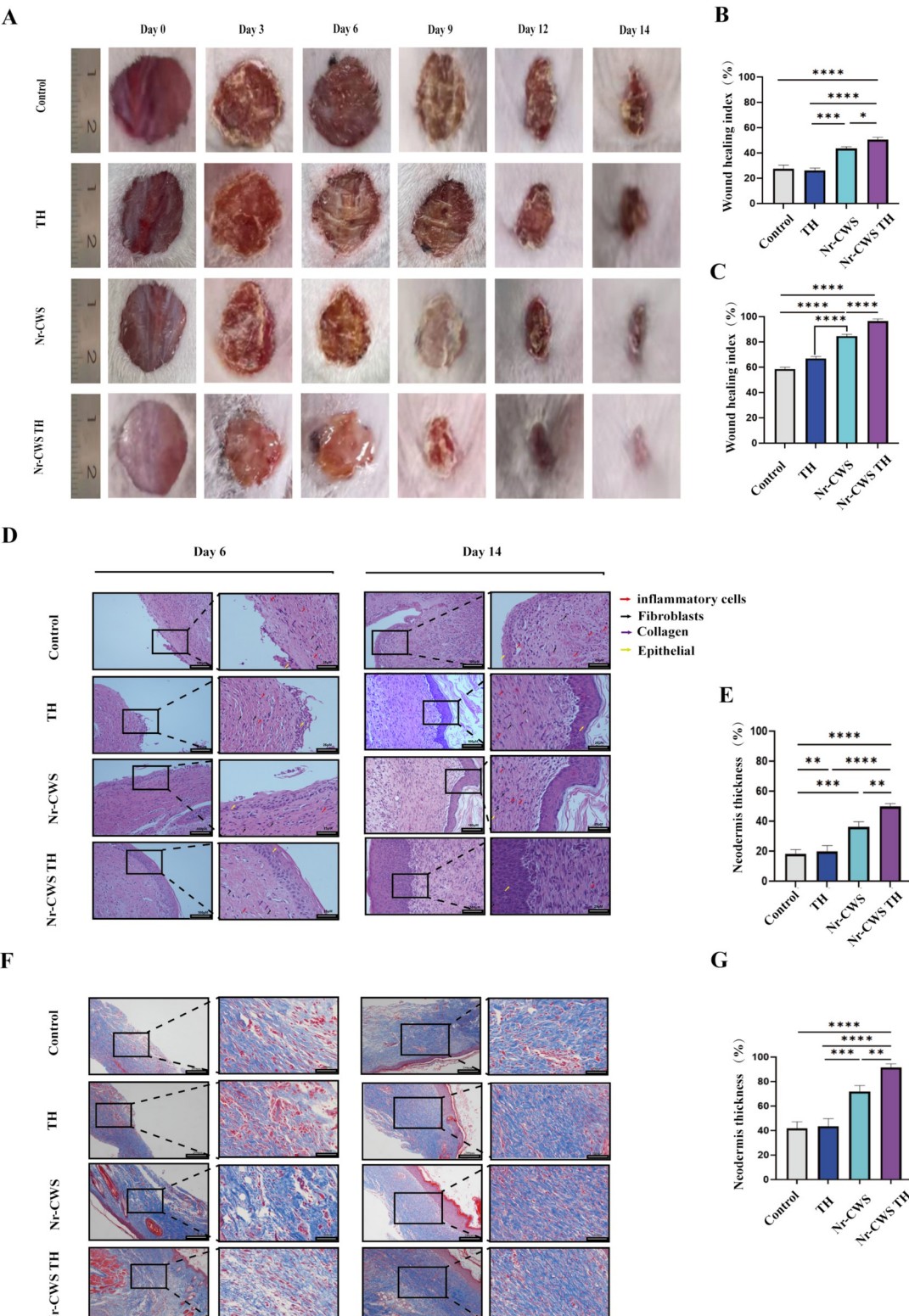

**Fig 4. Rate of wound closure.** The rate and condition of wound healing. A) The healing situation was observed from 0 to 14 day in different groups. B) Statistical analysis of the healing situation on the sixth day. C) Statistical analysis of the healing situation on the fourteenth day. D) On the sixth and 14th day, HE staining were performed to calculate the thickness of the newborn skin and statistical analysis of the thickness of the newborn skin. E) Statistical analysis of the thickness of the newborn

skin on the sixth day F) On the sixth and 14th day, Masson staining was performed. G) Statistical analysis of the thickness of the newborn skin on the 14th day.

was the highest, indicating that its local inflammation was well treated and the wound healing environment was the best.

The wound infected by diabetes has less angiogenesis at the wound site. Therefore, to evaluate wound angiogensis after hydrogel treatment, the well-known endothelial cell marker CD-31 was assessed. As shown in Fig 5A and 5B, CD-31 appeared in a large number of tissues in the Nr-CWS TH group. This finding implied the Nr-CWS TH group had a great trend of vascularization. The number of blood vessels in all groups on day 14 was higher than on that on day 6. The number in the control and TH groups increased the least, whereas that in the Nr-CWS TH group increased the most.

### 3.7 Therapeutic effects of Nr-CWS TH linked with activation of MAPK/ERK, PI3K/AKT, and JAK/STAT3 pathways

The whole layer of skin was cut and the protein was extracted to verify which factors are related to the healing of skin wounds. The VEGF and EGF contents in each experimental group were measured via Western blot. Their contents in the Nr-CWS TH group were the highest, followed by those in the Nr-CWS group. Phosphatidylinositol 3 kinase and protein kinase B (PI3K/Akt), Janus kinase/signal transducer and activator of transcription (JAK/STAT3), and mitogen-activated protein kinase kinase/extracellular signal-regulated kinase (MAPK/ERK) are three common signaling pathways that regulate cell proliferation. Western blot was used to detect the expression levels of p-AKT, AKT, p-ERK, ERK, p-STAT3, and STAT3 in epidermal tissue after 14 days of wound healing to reveal the potential mechanism of red card hydrogel promoting the proliferation and healing of epidermal cells. As shown in Fig 6, the expression levels of AKT, p-ERK, ERK, p-STAT3, and STAT3 in the Nr-CWS and Nr-CWS TH groups increased significantly compared with those in the control and blank hydrogel groups. The highest level of phosphorylation was found in the Nr-CWS group.

## 4 Discussion

Due to chronic inflammation and vascular injury, diabetic wound does not heal for a long time. Many doctors have made great efforts to solve this problem, but the current methods are limited [20]. In the present study, the healing effect of poloxamer hydrogel loaded with Nr-CWS on diabetic wound was explored. On the one hand, the effect of poloxamer hydrogel-loaded drugs was discussed, and the feasibility of applying them to wounds was verified. On the other hand, we studied the effect of Nr-CWS on diabetic wound was studied. The results showed that loading Nr-CWS onto Nr-CWS TH could effectively and safely promote wound healing in diabetic mice, and its mechanism may be related to the activation of MAPK/ERK, PI3K/AKT, and JAK/STAT3 signaling pathways.

Thermosensitive hydrogel plays an important role in the design of biomaterials due to its adjustable physical properties, controllable degradability, and easy fabrication [21, 22]. Poloxamer hydrogel is an ideal drug delivery tool compatible with most living tissues, facilitating its use in the treatment of regenerative disease and diabetes. A moist wound environment is well known to effectively promote cell proliferation and migration in addition to promoting resurfacing [23, 24]. High-thermosensitive-content hydrogels are viscoelastic and mechanically similar to biological tissues, and they provide a moist environment for wound beds. On the basis of the adjustable properties of thermosensitive materials, the optimal gel-sol transition state gel

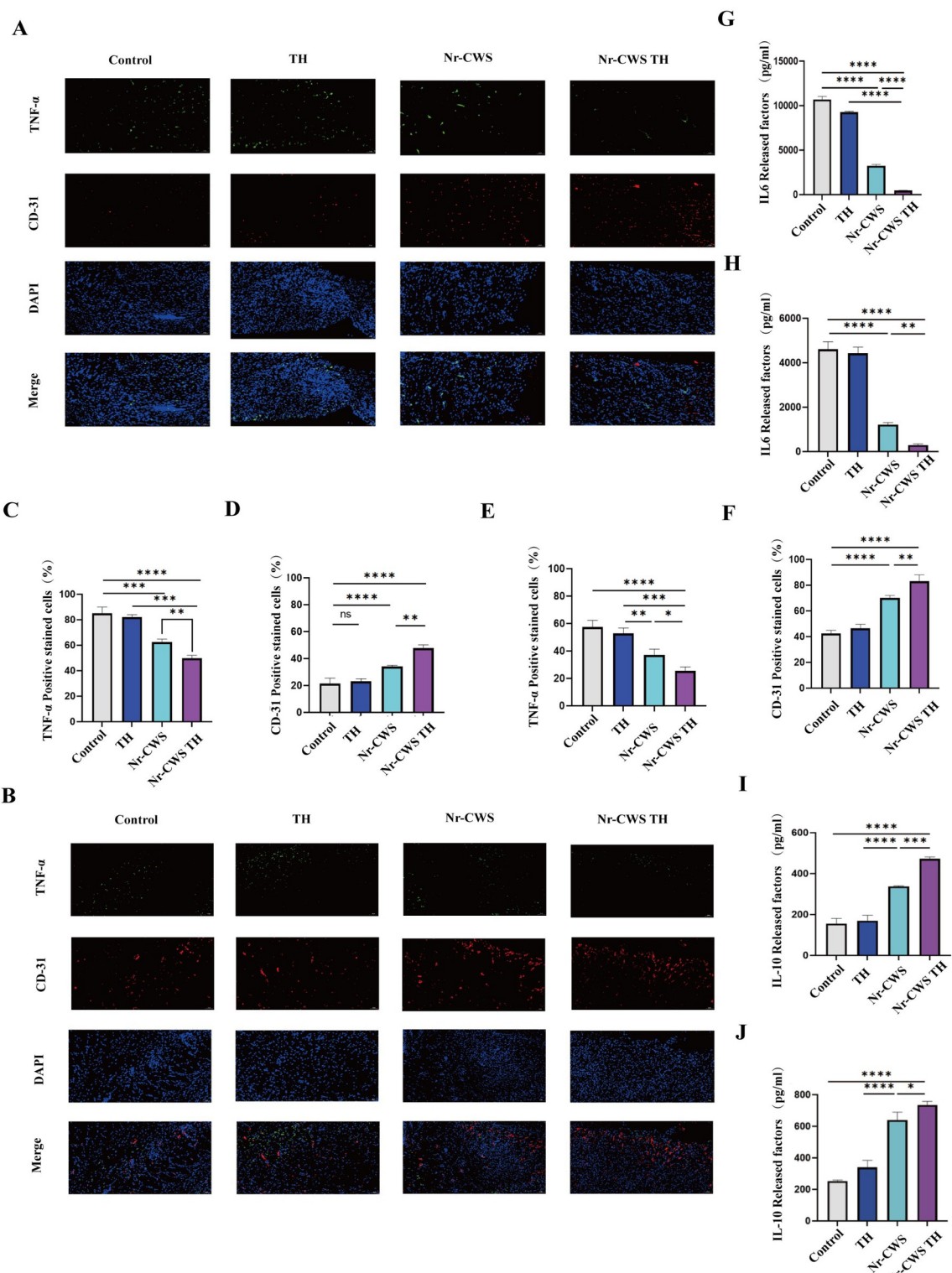

**Fig 5. The tissue fluorescence staining and Elisa of neonatal epidermis.** A, B) The neonatal epidermis was taken for tissue fluorescence staining, including TNF-α (green), CD31 (red), and cell nucleus (blue) on the sixth day (A) and on the fourteenth day (B). Scale bar = 50 μm. Scale bar = 50 μm. C,D) Statistical analysis of the TNF-αand CD-31 on the sixth day. E, F) Statistical analysis of the TNF-α and CD-31 on the fourteenth day. G, H) ELISA detected expression of IL6 in the wound healing tissue on sixth day (G) and 14th day (H). *p < 0.05, **p < 0.01, ***p < 0.001, ****p < 0.0001. I, J) ELISA detected expression of IL10 in the wound healing tissue on sixth day (E) and 14th day (F). *p < 0.05, **p < 0.01, ***p < 0.001, ****p < 0.0001.

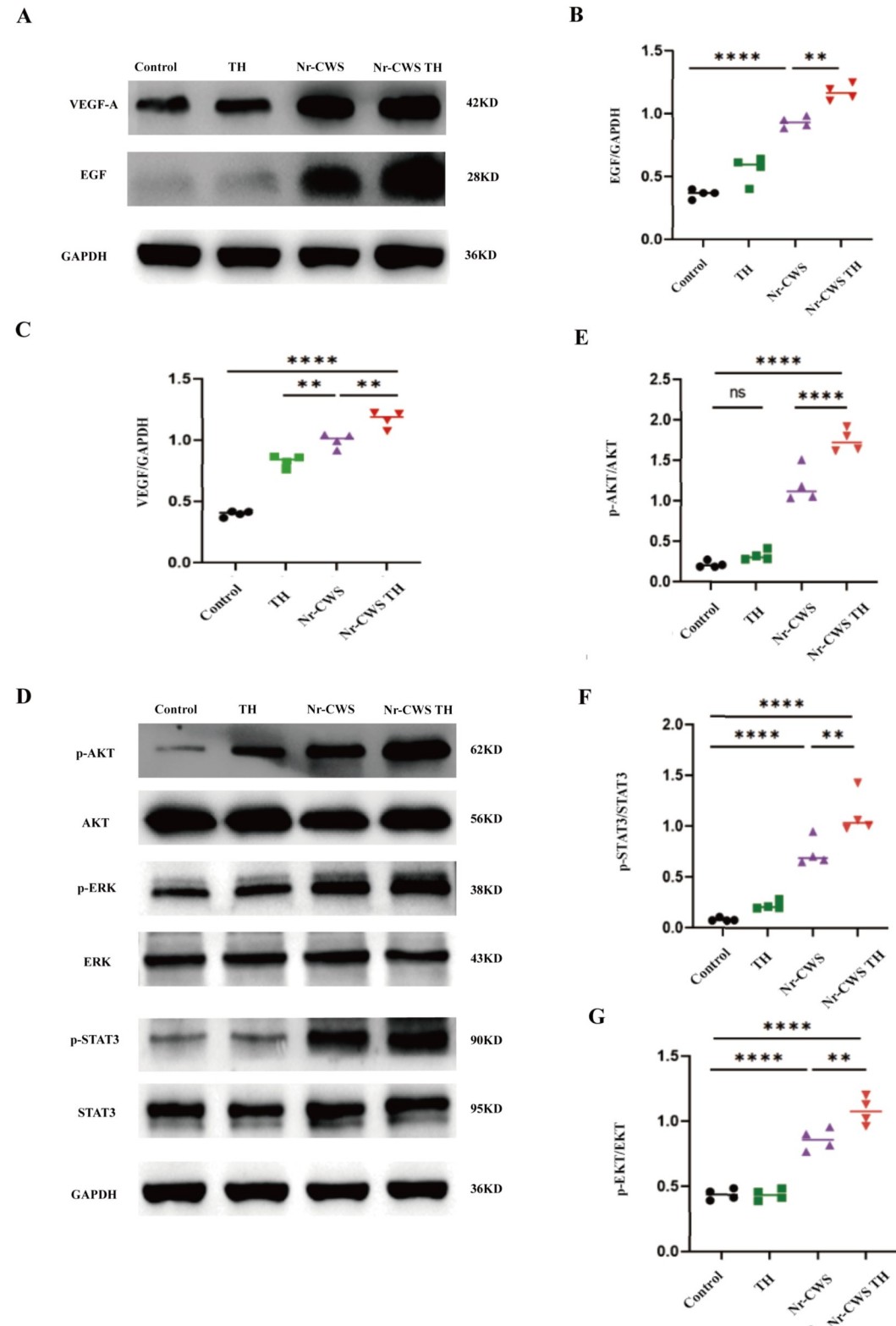

**Fig 6. Nr-CWS TH treatment promote wound healing through MAPK/ERK, PI3K/AKT and JAK/STAT3 pathways.** A) The protein levels of VEGF and EGF at 14 days by western blotting, GAPDH served as protein loading control); B, C) The statistical analysis of VEGF and EGF. The data are expressed in mean ±SEM, n = 4 for each group. $^{*}p < 0.05$, $^{**}p < 0.01$, $^{***}p < 0.001$, $^{****}p < 0.0001$. D) Immunoblot for p-AKT, p-ERK, p-STAT3; the proteins amount of AKT, ERK, STAT3

served as loading control; E, F, G) Densitometric statistical analysis of p-AKT/AKT, p-ERK/ERK and p-STAT3/STAT3, respectively. GAPDH was used as internal parameter. Data presented as mean ± SEM n = 4 for each group. *p < 0.05, **p < 0.01, ***p < 0.001, ****p < 0.0001.

was designed (Fig 1) in accordance with the response surface curve. A hydrogel containing P407/188 polymer and mono salivary ganglioside ganglioside (GM1 hydrogel) was found to be effective in preventing cell apoptosis and scar formation in rabbits [25]. High-glucose environment increases cell apoptosis and weakens the effect of drugs on the treatment of diabetic wounds. Therefore, P407/188 polymer was loaded onto Nr-CWS to combat high glucose and the poor chronic inflammation of diabetes. At present, poloxamer has proven its superiority in the administration of wounds, but it has not yet been evaluated for its application value in diabetic wounds. Therefore, in view of the slow release effect of poloxamer, Nr-CWS-loaded poloxamer hydrogel was used in this study to repair diabetic wound. The measurement of drug release showed that poloxamer extended the release time of Nr-CWS to more than 12 h. Therefore, it played a certain role in promoting and strengthening the curative effect. We found in the healing of diabetic wounds, and a large number of clinical cases made us further think that the large amount of secretion of diabetic wounds would prevent the healing of diabetic wounds. On the one hand, our thermosensitive material can form a gel shape to absorb secretions and form a natural protective layer within 12 hours of treatment. On the other hand, the drug can be released orderly within 12 hours. It's hard to do that with normal materials.

Nr-CWS could enhance the resistance to infection and stimulate the release of various cells to awaken various immune cells in the body. Excess TNF-α is detrimental to diabetic wound healing [26–28]. In the present study, Nr-CWS TH kept pro-inflammatory factors IL-6 and TNF-α at very low levels on days 6 and 14, thus accelerating would repair. Meanwhile, IL-10 has been shown to increase the secretion of anti-inflammatory factors, which, in turn, stimulate downstream cells for tissue repair. A study reported that in a diabetic wound animal model, N-CWS elevated revascularization via stimulating the expression of TGF-β1, which eventually accelerated wound healing. Moreover, immunohistochemistry and immunofluorescence data indicated that Nr-CWS TH treatment increased the quantity of new vessels and promoted wound healing. Nr-CWS has been used in animal models of diabetic wounds, and it was proven to be stimulated by TGF-1β. The expression of increased the activation of macrophages, promoted the reconstruction of blood circulation, and finally accelerated wound healing [19]. Fibroblasts are spindle-shaped speckled nuclei with two or more nucleoli surrounded by branched cytoplasm. Inflammatory cells are generally spherical and clumped together. Epidermal cells are the outermost structures with deep staining and dense coloring. The area where the fibroblasts are aggregated has a whole area of dark staining and it is where the collagen fibers are aggregated and multiplied. In HE staining, we found that the NR-CWSTH group had the thickest new skin thickness. Masson staining showed that the arrangement of fibroblasts was the most orderly among the four groups, and wound healing was the best. The present study confirmed that Nr-CWS could reduce IL-6 and TNF-α and increased the content of IL-10 in diabetic wounds to promote wound healing. As evidenced by the increase in CD-31, Nr-CWS could be concluded to have shown a strong angiogenesis effect. This experiment also has a limitation, that is, the mechanism of how Nr-CWS regulates the promotion and inhibition of inflammatory factors to a reasonable range in a short time is not clear.

The main biological role of Nr-CWS is very important for epidermal cell proliferation. Wound healing in diabetic wound mice was recently found to be related to the proliferation of MAPK/AKT, PI3K/AKT, and JAK/STAT3 signaling pathways. Jie and others have demonstrated that exocrine could activate the PI3K/AKT pathway to accelerate wound healing in

diabetes [29]. Magdalena has proven that MAPK/ERK is an important signaling pathway for endothelial cell migration and proliferation [30]. Leonard and others showed that JAK/STAT3 is a key part of many signaling pathways regulating cell growth, differentiation, survival, and pathogen resistance. This pathway involves the IL-6 (gp130) receptor family [31]. The phosphorylation of AKT, ERK, and STAT3 was studied by Western blot in the present paper to investigate whether Nr-CWS TH promotes wound healing by activating MAPK/AKT, PI3K/AKT and JAK/STAT3 signaling pathways. The in-vivo experiments showed that Nr-CWS treatment, especially when Nr-CWS was loaded on TH hydrogel, was significantly associated with increased phosphorylation of AKT, ERK, and STAT3 (Fig 5), indicating that these three signaling pathways may be activated after treatment. Through the above description of IL-6, JAK/STAT3 could be shown to play a certain role. The determination of CD-31, VEGF-A, and EGF could reveal that MAPK/AKT and PI3K/AKT pathways are activated in cell proliferation.

For the first time, P188 and P407 were used as carriers to load immune agents for the treatment of diabetic wounds. In terms of material production, on the one hand, we took into account the clinical feasibility, scientific, simple and easy operation, and on the other hand, we also took into account the possible problems in the process of full-layer wound healing to design experiments, and carried out an improved treatment plan through dressing change every 12h. In the production of materials, the viscosity and elastic coefficient of blank gelation slightly decreased, the process of sol-gel could be easily implemented. In general, this gelation process proved that Nr-CWS TH had an optimal phase transition temperature, which was especially suitable for diabetic wound healing on the surface of the body and for the drug to stay longer in the wound.

A notable detail that Nr-CWS TH hydrogel could accelerate wound healing faster than Nr-CWS solution, possibly because of the following: (1) hydrogel, as an appropriate scaffold, maximized the therapeutic potential of Nr-CWS, specifically targeted the expression of inflammatory factor presented in diabetic wounds, and promoted the accumulation of collagen in granulation tissue. (2) The continuous delivery of Nr-CWS had the least interference to the wound healing process. (3) gelation of Nr-CWS hydrogel on the wound bed could maintain the moist environment around the wound.

In summary, poloxamer hydrogels have a slow-release effect on the loaded drug to maximize contact between the drug and the wound. New other dressings were studied for the treatment of diabetic wounds. The exudates of adipose-derived stem cells or adipose derived mesenchymal stem cells were encapsulated in hydrogels, which significantly accelerated the closing of diabetic wounds [32, 33]. Despite these promising results, wound dressing using cells or cellular components still faces difficulty in clinical application. This composite poloxamer hydrogel has the advantages of low cost, convenient preparation, and easy adjustment. In addition, as an effective and safe drug delivery method, this hybrid hydrogel could apply any drug to skin wounds. The present study showed that thermosensitive epoxy propane ethylene oxide copolymer (thermosensitive poloxamer hydrogel) is a promising carrier. The poloxamer thermosensitive hydrogel loaded with Nr-CWS plays a good role in promoting angiogenesis and regulating inflammatory factors and the growth of epidermal cells. In addition, our mechanism showed that Nr-CWS could help activate MAPK/ERK, PI3K/AKT, and JAK/STAT3 signaling pathways, and promote wound healing. The use of Nr-CWS TH shortened the healing time of diabetic wounds, and the application of thermosensitive hydrogel provided additional ideas on drug loading and the treatment of diabetic wounds.

## 5 Conclusions

This study showed that smart thermosensitive poloxamer hydrogel is a promising carrier. Poloxamer loaded with Nr-CWS may activate MAPK/ERK, PI3K/Akt, and JAK/STAT3

signaling pathways to promote angiogenesis, and regulate inflammatory factors and epidermal cell growth.

## Supporting information

**S1 Fig. All western blots in the article.** The original images of western blots.
(TIF)

**S2 Fig. The biocompatibility of the hydrogel.** The comparison of wound skin stained with HE after using hydrogel.
(TIF)

## Author Contributions

**Conceptualization:** Jian Wang, Bingkun Zhao.

**Data curation:** Jian Wang, Bingkun Zhao, Liqun Jiang.

**Formal analysis:** Jian Wang, Bingkun Zhao.

**Funding acquisition:** Jian Wang, Bingkun Zhao.

**Investigation:** Jian Wang, Lili Sun.

**Methodology:** Jian Wang.

**Project administration:** Jian Wang.

**Resources:** Jian Wang, Qiang Li, Peisheng Jin.

**Validation:** Jian Wang, Peisheng Jin.

**Visualization:** Bingkun Zhao, Liqun Jiang, Qiang Li.

**Writing – original draft:** Jian Wang, Lili Sun.

**Writing – review & editing:** Jian Wang, Qiang Li, Peisheng Jin.

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
