## [Decision Letter · Decision Letter 0]

22 Aug 2022

PONE-D-22-22609Smart Thermosensitive Poloxamer Hydrogels Loaded with Nr-CWs for the Treatment of Diabetic WoundsPLOS ONE

Dear Dr. Jin,

Thank you for submitting your manuscript to PLOS ONE. After careful consideration, we feel that it has merit but does not fully meet PLOS ONE’s publication criteria as it currently stands. Therefore, we invite you to submit a revised version of the manuscript that addresses the points raised during the review process.

As you can see from the comments, the reviewers felt that the scientific soundness of this study should be improved before the acceptance of this work. I hope the specific comments from the reviewers will be useful for the major revision suggested by the reviewer. To help me expedite processing, please explicitly address the questions raised by the reviewers in your cover letter and also point out the changes made in the manuscript. I will go back to the reviewers for further input and advice before any final decision on possible publication is made.

We look forward to receiving your revised manuscript.

Kind regards,

Bing Xu, PhD

Academic Editor

PLOS ONE

Journal Requirements:

Reviewers' comments:

Reviewer's Responses to Questions

**Comments to the Author**

1. Is the manuscript technically sound, and do the data support the conclusions?

Reviewer #1: Partly

Reviewer #2: Partly

2. Has the statistical analysis been performed appropriately and rigorously? 

Reviewer #1: Yes

Reviewer #2: Yes

3. Have the authors made all data underlying the findings in their manuscript fully available?

Reviewer #1: Yes

Reviewer #2: Yes

4. Is the manuscript presented in an intelligible fashion and written in standard English?

Reviewer #1: No

Reviewer #2: No

5. Review Comments to the Author

Reviewer #1: Nocardia rubra cell wall skeleton (Nr-CWs) has been reported to accelerates wound healing. Authors here reported an improved wound healing property by delivering Nr-CWs with hydrogel. I find the results to be informative to a broad audience in drug delivery if the authors can address the following comments.

I was unable to find the figure legend in the submission and believe some of my concerns can be addressed by clarity in the legend.

1. How much Nr-CWs was loaded into the hydrogel delivery system? This is unclear throughout the manuscript. It would be important to quantify after preparing Nr-CWs in hydrogel, how much of Nr-CWs actually remain in the hydrogel as this determines the effective dosage.

2. In the cell viability plots generated by FACS, can author explicitly point out the time point when the data was taken post treatment?

3. Can authors clarify the relevance of HUVEC and FGF models for in vitro studies?

4. It is unclear to me the differences between Fig 2C vs 2D and Fig 2E and 2F.

5. To measure the release profile, authors labeled Nr-CWs with Cy5. Please add details of this process in the method section. Can authors be sure that measured release is Nr-CWs instead of Cy5 alone? Fig 3B should include Nr-CWs alone without hydrogel.

6. In Figure 3C, the release curve extended to day 14. Can the authors explain why retention of the Nr-CWs inside hydrogel improves wound healing? Doesn't it mean less released? What did I miss here.

7. "The release of hydrogel is slow released and has a good effect on the absorption of drugs." I found this statement unfounded here. Can authors comment on how absorption was measured? Related to last comment. If not, is this a hypothesis?

8. None of the groups in Figure 6 western blots were labeled.

Minor comment - abstract had missing parts of a sentence.

Reviewer #2: The manuscript by Wang et al. aims to access the effectiveness of a thermo-sensitive hydrogel supplemented with Nr-CWS in treating diabetic wounds. The authors claimed that this gel formulation is superior because it would reduce inflammation, increase angiogenesis and wound healing. The experiments are in general well designed and carried out, however, there are major concerns over the author’s writing style and the conclusion drawn regarding the intracellular cellular pathways related to the MOA of the gel.

1. The authors claim that the activation of PI3K/Akt, JAK/STAT3 and MAPK/ERK pathways explains the effectiveness of their hydrogel in promoting wound healing. However, two of these pathways, JAK/STAT3 and PI3K/Akt are also involved in inflammatory cytokine signaling transduction. For example, IL6 significantly up-regulates these pathways during tissue inflammation. Therefore, using these pathway activation as an indication of enhanced tissue healing is problematic.

2. There are no Figure Legend in this manuscript!!

3. The authors need to greatly improve their writing before the next submission. Currently there are too many grammatical errors throughout the manuscript and sentences that are phrased awkwardly, which make their statement hard to comprehend.

4. The authors need to improve their introduction. Currently, the authors mixed what they did in this manuscript and what has been discovered previously throughout the introduction section. Please make sure that you layout the currently issues of field, and proper background of hydrogel and Nr-CWS, SEPARATELY from your aim and what you have done in this paper.

5. Fig.4 needs to be improved or better explained. In 4D, the authors read out different cell type infiltrating the wound simply by looking at H&E staining. However, it is hard for the reader to follow what each arrows are pointing at. They need to explain what feature in H&E do you define as fibroblast etc, epithelial, etc. in 4F, the authors did PAS staining to look at collagen deposition. However, this methods was never mentioned in this manuscript. Additionally, collagen deposition is not always related to good healing, it rather indicate tissue fibrosis in many cases. The authors need to explain what they are looking at in these PAS staining images and why it helps support their claims.

Minor concerns:

1. The figures need to be called in sequence. For example, Fig. 2B is called in section 3.2 before Fig. 2A. In this case, you may want to change the order of the two figures or change the order of how you discuss your data. Same issue in Fig. 3.

2. How is Fig. 2C and 2D different?

3. Authors need to better describe their methods. For example, in Fig. 2, they used FACS to determine cell apoptosis by staining Annexin. However, they did not provide enough explanation on what they measured and why they used is as the marker for apoptosis. Readers who are not familiar with this method would not understand this experiment. Please provide more information.

6. PLOS authors have the option to publish the peer review history of their article (what does this mean?). If published, this will include your full peer review and any attached files.

Reviewer #1: No

Reviewer #2: No

---

## [Author Response · Author response to Decision Letter 0]

20 Sep 2022

Reviewer #1: Nocardia rubra cell wall skeleton (Nr-CWs) has been reported to accelerates wound healing. Authors here reported an improved wound healing property by delivering Nr-CWs with hydrogel. I find the results to be informative to a broad audience in drug delivery if the authors can address the following comments.

I was unable to find the figure legend in the submission and believe some of my concerns can be addressed by clarity in the legend.

Response: It's a great honor to get your affirmation which makes our whole team very excited and everyone is full of fighting spirit. We revised the manuscript based on the questions you raised. 

1. How much Nr-CWs was loaded into the hydrogel delivery system? This is unclear throughout the manuscript. It would be important to quantify after preparing Nr-CWs in hydrogel, how much of Nr-CWs actually remain in the hydrogel as this determines the effective dosage.

Response: In this experiment, 60μg Nr-CWS was added per milliliter of the hydrogel delivery system, which was based on the drug concentrate on used clinically. The aqueous preparation of nr-cws was also at a concentration of 60ug/ml. In figure3B and C, we measured the drug release in gel under different conditions in vitro and in vivo. These values reflect the effective dose. We put this change in line 110 and 153-154.

2. In the cell viability plots generated by FACS, can author explicitly point out the time point when the data was taken post treatment?

Response: Flow cytometry will be performed with two cell types 12 hours after drug and gel treatment. The reason we chose 12h is because we use 12h per dose. 12h processing can better obtain the data closest to ordinary use. It is more appropriate for clinical use. This change is listed in Flow Cytometry section Line 131 and Results section Line 241.

3. Can authors clarify the relevance of HUVEC and FGF models for in vitro studies?

Response: Huvec and Fibroblast play a very important role in wound healing. We more visually verified that the drugs and materials did not cause cell apoptosis by flow cytometry. There was no toxic effect on cells

4. It is unclear to me the differences between Fig 2C vs 2D and Fig 2E and 2F.

Response: Figure 2C is the flow chart of HUVEC cells, and E is the corresponding statistical chart. Figure 2D is the flow chart of Fibroblast, and F is the corresponding statistical chart. In this part, we have re-uploaded the figure legend, which has a more detailed description.

5. To measure the release profile, authors labeled Nr-CWs with Cy5. Please add details of this process in the method section. Can authors be sure that measured release is Nr-CWs instead of Cy5 alone? Fig 3B should include Nr-CWs alone without hydrogel.

Response: Nr-CWS is a suspension that is insoluble in water. If we want to test the release of Nr-CWS, we need to mark its functional group with a marker. Finally we used CY5 to tag this drug. We describe this part of details in line160-166. The functional group of the Cy5 dye is covalently linked to the amino group of Nr-CWS. This covalent attachment is stable in vitro so we measured Nr-CWS instead of Cy5.

6. In Figure 3C, the release curve extended to day 14. Can the authors explain why retention of the Nr-CWs inside hydrogel improves wound healing? Doesn't it mean less released? What did I miss here.

Response: Although we release it slowly, the gel can prolong the retention of Nr-CWS on the skin. The contact time between the Nr-CWS and the skin is longer, which can prolong the action time of the drug on the local skin.

7. "The release of hydrogel is slow released and has a good effect on the absorption of drugs." I found this statement unfounded here. Can authors comment on how absorption was measured? Related to last comment. If not, is this a hypothesis?

Response: We are taking this approach in order to prolong the local effect of the drug and a more convenient and efficient way to use a drug in a specific location for a longer time. Our sentence is not rigorous enough, so we have revised this part in line 255

8. None of the groups in Figure 6 western blots were labeled.

Response: It has been updated as requested. Thank you for your advice.

Minor comment - abstract had missing parts of a sentence.

Response: Thanks for your instructions, we have proofread the whole article for many times. Grammar and expression problems are solved. Thank you again for your valuable advice.

Reviewer #2: The manuscript by Wang et al. aims to access the effectiveness of a thermo-sensitive hydrogel supplemented with Nr-CWS in treating diabetic wounds. The authors claimed that this gel formulation is superior because it would reduce inflammation, increase angiogenesis and wound healing. The experiments are in general well designed and carried out, however, there are major concerns over the author’s writing style and the conclusion drawn regarding the intracellular cellular pathways related to the MOA of the gel.

Response: Thank you for your affirmation of this paper and this experiment. We have tried our best to answer all your questions. We hope that the explanation of the pathway can satisfy you. The study of pathways has always been the key to medical research. We found two articles in PLOS ONE journal and Biomaterials Journal about the effect of IL-6 on the pathway and the validation of this signaling pathway after tissue damage in diabetic wound mice. Hope to be able to answer your questions.

1. The authors claim that the activation of PI3K/Akt, JAK/STAT3 and MAPK/ERK pathways explains the effectiveness of their hydrogel in promoting wound healing. However, two of these pathways, JAK/STAT3 and PI3K/Akt are also involved in inflammatory cytokine signaling transduction. For example, IL6 significantly up-regulates these pathways during tissue inflammation. Therefore, using these pathway activation as an indication of enhanced tissue healing is problematic.

Response: I think you are right about the information you mentioned about the JAK/STAT3 and PI3K/Akt pathways. In the case of normal wounds, the increase of proinflammatory factor IL-6 will lead to the increase of JAK/STAT3 pathway related protein expression, and then affect the relevant healing. However, we looked at the literature and noticed that during chronic wound healing, the JAK/STAT pathway needs to be upregulated when its normal function is impaired, especially in the context of senescent cells and reduced growth factor/receptor potential. We have noticed that the wounds of diabetic patients are very complex. Compared with ordinary wounds, JAK/STAT3 pathway needs more activation to ensure subsequent healing. Wound repair experiments in drosophila have shown that JAK/STAT signaling works in tandem with Wingless (Wg) signaling to activate and promote the proliferation of regenerative cells. Beigel et al.(1) showed a time-dependent induction of STAT1 and STAT3 tyrosine phosphorylation in injured intestinal epithelial cells, which reached a maximum value after 10-30 min. They also found a two-fold increase in STAT1 and STAT3, a 1.5-fold increase in STAT6, and a 16-fold increase in STAT5B. In the same experimental model, the up-regulation of STAT and activation of STAT protein were directly associated with wound healing in intestinal epithelial cells. At the same time, we also noted that Rui Li et al.(2) studied the JAK/STAT3 and PI3K/Akt pathways in the neural tissues of diabetic mice and verified their role in tissue repair. Therefore, in combination with the above literature, we believe that IL-6 may have an effect on the pathway, but it has little effect on this pathway during the wound repair period. Moreover, there is no positive correlation between IL-6 and the protein activation content of the pathway at 14 days, so our research group believes that these two pathways can also be used as indicators for repair evaluation.

2. There are no Figure Legend in this manuscript!!

Response: We are very sorry that Figure Legend cannot be uploaded successfully due to our operation error. I would like to apologize.

3. The authors need to greatly improve their writing before the next submission. Currently there are too many grammatical errors throughout the manuscript and sentences that are phrased awkwardly, which make their statement hard to comprehend.

Response: Our team further refined and polished the paper. Thank you for your questions to make our article more rigorous.

4. The authors need to improve their introduction. Currently, the authors mixed what they did in this manuscript and what has been discovered previously throughout the introduction section. Please make sure that you layout the currently issues of field, and proper background of hydrogel and Nr-CWS, SEPARATELY from your aim and what you have done in this pa per.

Response: We re-wrote the background and introduction separately. We improved our introduction. Thank you for your warning

5. Fig.4 needs to be improved or better explained. In 4D, the authors read out different cell type infiltrating the wound simply by looking at H&E staining. However, it is hard for the reader to follow what each arrows are pointing at. They need to explain what feature in H&E do you define as fibroblast etc, epithelial, etc. in 4F, the authors did PAS staining to look at collagen deposition. However, this methods was never mentioned in this manuscript. Additionally, collagen deposition is not always related to good healing, it rather indicate tissue fibrosis in many cases. The authors need to explain what they are looking at in these PAS staining images and why it helps support their claims.

Response: Various cells were described in detail, and the results of Masson staining were judged by the degree of disorder in the arrangement of fibroblasts. If this part is hyperplastic due to collagen deposition, the arrangement of its fibroblasts will be disordered, which can identify whether the wound is in good healing condition. We have strictly added line to this part Line 278-280 and line 371-374

Minor concerns:

1. The figures need to be called in sequence. For example, Fig. 2B is called in section 3.2 before Fig. 2A. In this case, you may want to change the order of the two figures or change the order of how you discuss your data. Same issue in Fig. 3.

Response: Thank you for your advice. We have changed the position. Line238，240，251 and 254

2. How is Fig. 2C and 2D different?

Response: Figure 2C is the flow chart of HUVEC cells, and E is the corresponding statistical chart. Figure 2D is the flow chart of Fibroblast, and F is the corresponding statistical chart. In this part, we have re-uploaded the figure legend, which has a more detailed description.

3. Authors need to better describe their methods. For example, in Fig. 2, they used FACS to determine cell apoptosis by staining Annexin. However, they did not provide enough explanation on what they measured and why they used is as the marker for apoptosis. Readers who are not familiar with this method would not understand this experiment. Please provide more information.

Response: The Annexin V-FITC Apoptosis Detection Kit is a cell Apoptosis Detection Kit that uses FITC-labeled recombinant human Annexin V to detect phosphatidylserine present on the cell membrane surface during Apoptosis. Phosphatidyl serine (PS) is located on the cytosolic side of the cell membrane. After early apoptosis, PS everted to the cell membrane side, and Annexin V had a high affinity for PS. When Annexin V was labeled with FITC, cells with PS everted emitted green fluorescence due to Annexin V-FITC binding. In the late stage of apoptosis, the cell membrane structure is damaged, propidium iodide (PI) enters the cell membrane through the cell membrane, and the cell emits red and green fluorescence at the same time. Annexin V-FITC (green fluorescence) and PI (red fluorescence) signals were detected at the same time, and the proportion of early apoptotic cells and late apoptotic cells was analyzed. The detailed operation steps are added in Line 131-136. 

1. Beigel, F., Friedrich, M., Probst, C., Sotlar, K., Goke, B., Diegelmann, J., and Brand, S. (2014) Oncostatin M mediates STAT3-dependent intestinal epithelial restitution via increased cell proliferation, decreased apoptosis and upregulation of SERPIN family members. PLoS One 9, e93498

2. Li, R., Li, Y., Wu, Y., Zhao, Y., Chen, H., Yuan, Y., Xu, K., Zhang, H., Lu, Y., Wang, J., Li, X., Jia, X., and Xiao, J. (2018) Heparin-Poloxamer Thermosensitive Hydrogel Loaded with bFGF and NGF Enhances Peripheral Nerve Regeneration in Diabetic Rats. Biomaterials 168, 24-37

---

## [Decision Letter · Decision Letter 1]

17 Oct 2022

PONE-D-22-22609R1Smart Thermosensitive Poloxamer Hydrogels Loaded with Nr-CWs for the Treatment of Diabetic WoundsPLOS ONE

Dear Dr. Jin,

Thank you for submitting your manuscript to PLOS ONE. After careful consideration, we feel that it has merit but does not fully meet PLOS ONE’s publication criteria as it currently stands. Therefore, we invite you to submit a revised version of the manuscript that addresses the points raised during the review process.

As you can see from the enclosed reviews, the reviewers find your manuscript potentially suitable for publication in PLoS One. However, some of them raised a few specific issues that must be addressed before the final acceptance of your manuscript. Hence, I am requesting that you submit a revised version of this manuscript to address the comments. To help me expedite processing, please explicitly address the questions raised by the reviewer in your cover letter and point out the changes made in the manuscript.

We look forward to receiving your revised manuscript.

Kind regards,

Bing Xu, PhD

Academic Editor

PLOS ONE

Journal Requirements:

Reviewers' comments:

Reviewer's Responses to Questions

**Comments to the Author**

1. If the authors have adequately addressed your comments raised in a previous round of review and you feel that this manuscript is now acceptable for publication, you may indicate that here to bypass the “Comments to the Author” section, enter your conflict of interest statement in the “Confidential to Editor” section, and submit your "Accept" recommendation.

Reviewer #1: (No Response)

Reviewer #3: All comments have been addressed

Reviewer #4: (No Response)

Reviewer #5: All comments have been addressed

2. Is the manuscript technically sound, and do the data support the conclusions?

Reviewer #1: Yes

Reviewer #3: Yes

Reviewer #4: (No Response)

Reviewer #5: Partly

3. Has the statistical analysis been performed appropriately and rigorously? 

Reviewer #1: Yes

Reviewer #3: Yes

Reviewer #4: (No Response)

Reviewer #5: N/A

4. Have the authors made all data underlying the findings in their manuscript fully available?

Reviewer #1: Yes

Reviewer #3: Yes

Reviewer #4: (No Response)

Reviewer #5: No

5. Is the manuscript presented in an intelligible fashion and written in standard English?

Reviewer #1: Yes

Reviewer #3: Yes

Reviewer #4: (No Response)

Reviewer #5: Yes

6. Review Comments to the Author

Reviewer #1: In method section regarding release profile of Nr-CWS from hydrogel in vivo, authors still did not clarify the chemistry used for the cy5 conjugation. Is this NHS chemistry? Please add this information into your method beyond 'covalent'. How is the free dye removed and validated? This is critical information in deciding the validity of all of the drug release data. In addition, how does Cy5 disrupts this protein treatment if the modification is covalent? Another way of asking this question would be, is the Nr-CWS hydrogel treatment even more effective without Cy5 labeling?

Reviewer #3: (No Response)

Reviewer #4: In this manuscript, the author reported hydrogel loading Nocardia rubra cell wall skeleton (Nr-CWs) to accelerates wound healing. This hydrogel would reduce inflammation, increase angiogenesis and wound healing. In general, this work is well designed and carried out. However, there also have some minor issues need to be revised:

1. Why did the author choose P407 and P188 as the composition of hydrogels? The degradation of hydrogels in vivo needs to be confirmed.

2. Please confirm that Nr-CWS was successfully loaded inside the hydrogel.

3. The biocompatibility of the hydrogel needs to be confirmed before in vivo animal experiments.

4. Diabetic wound healing is a complex and continuous process. The manuscript indicates that Nr-CWS in the hydrogel releases about 60% of the original within 12 h. This does not seem to meet the need for sustained wound repair. Once the release of Nr-CWS in the hydrogel is terminated, how the hydrogel promotes wound repair?

5. Whether collagen deposition at the wound site after hydrogel treatment causes excessive fibrosis of the wound, explain them.

6. Why did the author not continue the drug release test for Nr-CWS release performance?

7. In the manuscript, the experimental results are described excessively, and the discussion of scientific problems is lacking. The author had better analyze the scientific problems in the manuscript in detail. In addition, the novelty of this work still needs to be highlighted in the introduction and conclusion.

Reviewer #5: In manuscript ‘Smart Thermosensitive Poloxamer Hydrogels Loaded with Nr-CWs for the Treatment of Diabetic Wounds’, the authors designed a P188/P407 poloxamers hydrogel mixed with Nr-CWS to promote angiogenesis and wound repair. This paper will be of interest in readers working in the field of Full-thickness skin defect repair. Some revisions are required to make it appropriate for publication, and the issues related to these revisions are listing below:

Q1. There are not enough data to explain the characteristics and innovations of this study.

Q2 In ‘The slow release prolongs the duration of the drug's effect on the wound in vitro and in vivo’ why did only test the drug’s release in 12 hours? More time points should be tested.

Q3. We found no description of the degradation performance of poloxamers hydrogels.

Q4. In ‘Fig.4D’, we didn’t see the scale bar in both 6 and 14 days.

Q5. In ‘Fig.4F’, we can hardly see the difference in collagen deposition between four groups in 14 days.

Q6. There were still format problems. For example, the title ‘Western blot analysis’ should be bold. ‘Establishment of Diabetic diabetic wound model’ should be deleted ‘Diabetic’; the ‘°C’ in line 106 and 159 were different and we didn’t see a space between ‘0.5h,shaken’ in line 160 so on.

7. PLOS authors have the option to publish the peer review history of their article (what does this mean?). If published, this will include your full peer review and any attached files.

Reviewer #1: No

Reviewer #3: No

Reviewer #4: No

Reviewer #5: No

---

## [Author Response · Author response to Decision Letter 1]

28 Nov 2022

Dear Editors and Reviewers:

Thanks for your letter and for the reviewers' comments concerning our manuscript entitled “Smart thermosensitive poloxamer hydrogels loaded with Nr-CWs for the treatment of diabetic wounds”. Those comments are valuable for revising and improving our paper with important guiding significance. We have made correction according to the comments, revised portion are highlighted in the paper. The responds to the reviewers' comments are as follows:

Reviewer 1：

Reviewer #1: In method section regarding release profile of Nr-CWS from hydrogel in vivo, authors still did not clarify the chemistry used for the cy5 conjugation. Is this NHS chemistry? Please add this information into your method beyond 'covalent'. How is the free dye removed and validated? This is critical information in deciding the validity of all of the drug release data. In addition, how does Cy5 disrupts this protein treatment if the modification is covalent? Another way of asking this question would be, is the Nr-CWS hydrogel treatment even more effective without Cy5 labeling?

Response：We are very sorry that we did not add the detailed explanation to the experimental part. As you said, this reaction was made by NHS chemistry. We used the cy5 marker only to measure its release, not to add cy5 to the drug. We finally tested the absence of excess cy5 fluorescent dye in the filtrate through dialysis bags with 8000 molecular weight. Indicating that no excess fluorescent dye affects the final result. We place this revision at: （Line 171-174）.

Reviewer #3: (No Response)

Reviewer #4: In this manuscript, the author reported hydrogel loading Nocardia rubra cell wall skeleton (Nr-CWs) to accelerates wound healing. This hydrogel would reduce inflammation, increase angiogenesis and wound healing. In general, this work is well designed and carried out. However, there also have some minor issues need to be revised:

Response:Thank you very much for your affirmation of our research group. The members of our research group have carefully revised the questions you raised and believe that your guidance has played a decisive role in the improvement of our article. After careful research, we will respond to your questions one by one. Hope to get your approval.

1.Why did the author choose P407 and P188 as the composition of hydrogels? The degradation of hydrogels in vivo needs to be confirmed.

Response：Neither P407 nor P188 can reach the human body temperature and just become gel state, so the transition temperature can be controlled through matching. As for the first confirmation, we explained it from the following three aspects: First, we reviewed the literature and found that either P188 or P407 alone would not have an impact on organisms (1, 2). Secondly, we use it on the surface of the wound surface. It is more difficult for large molecules to enter the body and participate in the circulation than small molecules. Finally, we also injected the mice subcutaneously. After dissolution, we sliced the skin of the mice and observed that no other discomfort was found, so we chose the mixed material to treat the wound. This part of the supplement is modified at:(Line73-76)

2.Please confirm that Nr-CWS was successfully loaded inside the hydrogel.

Response：Sorry we didn't explain that clearly. First of all, we conducted comprehensive detection of the blank gel and the gel coated with the drug in Fig1. We found that the drug Nr-CWS had been encapsulated in the pores by electron microscopy, which was also verified by subsequent release experiments. Therefore, we believe that Nr-CWS was successfully incorporated into the gel. The location is marked (Line 246-247)

3. The biocompatibility of the hydrogel needs to be confirmed before in vivo animal experiments.

Response：As mentioned earlier, we have studied biocompatibility from three aspects. Firstly, we read the literature and found that both P407 and P188 alone have been demonstrated in new aspects of biocompatibility, which will not affect normal cell tissues (3, 4). Secondly, we tried to find the potential danger by subcutaneous injection of mice, but we did not find that these macromolecules would affect normal tissues by observing the HE staining of skin wounds (FigS2). Thirdly, we believe that when gel with drugs is applied to the wound surface, macromolecular gel is more difficult to enter the body than small molecular substances, so the impact will be smaller. And after 12h, we will clean the residual gel. From the above point of view, we believe that its biocompatibility has been well certified. This revision is placed on Line: (Line73-76)

4.Diabetic wound healing is a complex and continuous process. The manuscript indicates that Nr-CWS in the hydrogel releases about 60% of the original within 12 h. This does not seem to meet the need for sustained wound repair. Once the release of Nr-CWS in the hydrogel is terminated, how the hydrogel promotes wound repair?

Response：We apologize for the misunderstanding caused by our rough explanation. As is known to all, in the treatment of diabetic wounds, the wound microenvironment is very poor. In the long treatment cycle, we must carry out debridement regularly to ensure a clean and tidy wound. Therefore, we found that a clinical cycle of 12h is the most effective treatment plan, and a 12h wound cleaning has a good effect on both the patient and the wound healing. Therefore, this gel is also released once for 12h, during which the gel will not only release the drug but also provide good protection to the wound. However, the delivery environment of diabetic wound and gel in the body is different. Diabetic wound will not only have a lot of exudates, but also contact with outside air will cause harmful substances to affect wound healing. Therefore, this gel can absorb exudates and physically block harmful substances from the outside to accelerate wound healing and wound repair.We put the explanation in (Line365-370) of the article.

5.Whether collagen deposition at the wound site after hydrogel treatment causes excessive fibrosis of the wound, explain them.

Response:The question you raised is also the problem we met in our original study. Whether collagen fibers are hyperplasia is mainly to observe whether the arrangement of collagen fibers is disordered. When the staining is dark and the arrangement of collagen fibers is chaotic, we will consider this part as collagen fiber hyperplasia. Through the experiment, we can find that the growth and arrangement of collagen fibers are very regular after the use of gel, so we can conclude that there is no problem of excessive fiber hyperplasia.

6.Why did the author not continue the drug release test for Nr-CWS release performance?

Response:Thank you very much for your question, as pointed out above. Due to the particularity of the diabetic environment, wound cleaning is required regularly to ensure the best effect. However, our drug use time is just once 12h administration frequency, so we did not continue to evaluate the release performance of the drug.

7.In the manuscript, the experimental results are described excessively, and the discussion of scientific problems is lacking. The author had better analyze the scientific problems in the manuscript in detail. In addition, the novelty of this work still needs to be highlighted in the introduction and conclusion.

Response:Our team is very sorry for the confusion in the reading. Here, our team has carefully deleted and supplemented these parts, hoping to get your approval.Lin97-103 and Line384-388 and Line 416-426.

Reviewer #5: In manuscript ‘Smart Thermosensitive Poloxamer Hydrogels Loaded with Nr-CWs for the Treatment of Diabetic Wounds’, the authors designed a P188/P407 poloxamers hydrogel mixed with Nr-CWS to promote angiogenesis and wound repair. This paper will be of interest in readers working in the field of Full-thickness skin defect repair. Some revisions are required to make it appropriate for publication, and the issues related to these revisions are listing below:

Response:Thank you very much for your recognition of our team's work. The whole team is very encouraged to receive your comments. We believe that there is still a long way to go in the treatment of diabetes wounds, but we hope that our work can give some guidance to future generations. Hope to solve the problem of diabetic wound healing as soon as possible.

Q1. There are not enough data to explain the characteristics and innovations of this study.

Response:We are very sorry that we did not emphasize this point in the course of the article writing and the experiment. We will write the introduction and discussion section again. I have focused on replying to your question. We hope that your approval of our supplementary deletion can be obtained, and this article will be published in plos one.Lin97-103 and Line384-388 and Line 416-426.

Q2 In ‘The slow release prolongs the duration of the drug's effect on the wound in vitro and in vivo’ why did only test the drug’s release in 12 hours? More time points should be tested.

Response:We have this problem in our research. The most true state is that complex diabetic wounds are associated with exudation and external physical stimulation on a daily basis, resulting in slow healing problems. However, no matter which preparation of material is applied on the limited wound surface, it is impossible to completely solve these problems. So dressing change debridement has become a problem to be solved at this stage. In the long process of wound healing. Wounds cannot be solved with a single application. So we noticed this in the clinical transformation process and came up with this gel.

Q3. We found no description of the degradation performance of poloxamers hydrogels.

Response:The degradation performance of this gel is beyond doubt, on the one hand, single P188 and P407 have been proven (1,2). Secondly, it is difficult for macromolecular gels to enter the body from skin wounds, and we have proved in cell experiments that they do not have toxic effects on cells. Therefore, we believe that this gel is safe and reliable. In the use cycle once every 12h. The main function of our macromolecular gel is still wound protection and drug release.we did not find that these macromolecules would affect normal tissues by observing the HE staining of skin wounds (FigS2). So it's almost never metabolized in the body and it doesn't affect safety.I'm sorry, we have explained this problem. Put the modified part in Line (Line73-76)

Q4. In ‘Fig.4D’, we didn’t see the scale bar in both 6 and 14 days

Response:I am very sorry that we ignored this problem in our previous work. We have revised it and uploaded it again.

Q5. In ‘Fig.4F’, we can hardly see the difference in collagen deposition between four groups in 14 days.

Response:We can tell by the color and arrangement of the collagen fibers. We found that the structure arrangement of Nr-CWS gel was more orderly than that of blank gel. That means it heals better. However, the color of diabetic wounds was significantly lighter than that of blank gel, indicating that the density and growth of collagen fibers were far lower than that of gel group. We believe that the repair of collagen fibers is quite obvious in this comparison.

Q6. There were still format problems. For example, the title ‘Western blot analysis’ should be bold. ‘Establishment of Diabetic diabetic wound model’ should be deleted ‘Diabetic’; the ‘°C’ in line 106 and 159 were different and we didn’t see a space between ‘0.5h,shaken’ in line 160 so on.

Response:I'm really sorry that we have rearranged and modified all the parts.We make all °C in the text the same See in Line 116, 117, 147 and 171 and so on.

Reference

1. Tan MSA, Pandey P, Lohman RJ, Falconer JR, Siskind DJ, Parekh HS. Fabrication and Characterization of Clozapine Nanoemulsion Sol-Gel for Intranasal Administration. Mol Pharm. 2022;19(11):4055-66.

2. Kjar A, Wadsworth I, Vargis E, Britt DW. Poloxamer 188 - quercetin formulations amplify in vitro ganciclovir antiviral activity against cytomegalovirus. Antiviral Res. 2022;204:105362.

3. Moiseev RV, Steele F, Khutoryanskiy VV. Polyaphron Formulations Stabilised with Different Water-Soluble Polymers for Ocular Drug Delivery. Pharmaceutics. 2022;14(5).

4. Kamel R, El-Wakil NA, Elkasabgy NA. Calcium-Enriched Nanofibrillated Cellulose/Poloxamer in-situ Forming Hydrogel Scaffolds as a Controlled Delivery System of Raloxifene HCl for Bone Engineering. Int J Nanomedicine. 2021;16:6807-24.

Dear Editors and Reviewers:

Thanks for your letter and for the reviewers' comments concerning our manuscript entitled “Smart thermosensitive poloxamer hydrogels loaded with Nr-CWs for the treatment of diabetic wounds”. Those comments are valuable for revising and improving our paper with important guiding significance. We have made correction according to the comments, revised portion are highlighted in the paper. The responds to the reviewers' comments are as follows:

Reviewer 1：

Reviewer #1: In method section regarding release profile of Nr-CWS from hydrogel in vivo, authors still did not clarify the chemistry used for the cy5 conjugation. Is this NHS chemistry? Please add this information into your method beyond 'covalent'. How is the free dye removed and validated? This is critical information in deciding the validity of all of the drug release data. In addition, how does Cy5 disrupts this protein treatment if the modification is covalent? Another way of asking this question would be, is the Nr-CWS hydrogel treatment even more effective without Cy5 labeling?

Response：We are very sorry that we did not add the detailed explanation to the experimental part. As you said, this reaction was made by NHS chemistry. We used the cy5 marker only to measure its release, not to add cy5 to the drug. We finally tested the absence of excess cy5 fluorescent dye in the filtrate through dialysis bags with 8000 molecular weight. Indicating that no excess fluorescent dye affects the final result. We place this revision at: （Line 171-174）.

Reviewer #3: (No Response)

Reviewer #4: In this manuscript, the author reported hydrogel loading Nocardia rubra cell wall skeleton (Nr-CWs) to accelerates wound healing. This hydrogel would reduce inflammation, increase angiogenesis and wound healing. In general, this work is well designed and carried out. However, there also have some minor issues need to be revised:

Response:Thank you very much for your affirmation of our research group. The members of our research group have carefully revised the questions you raised and believe that your guidance has played a decisive role in the improvement of our article. After careful research, we will respond to your questions one by one. Hope to get your approval.

1.Why did the author choose P407 and P188 as the composition of hydrogels? The degradation of hydrogels in vivo needs to be confirmed.

Response：Neither P407 nor P188 can reach the human body temperature and just become gel state, so the transition temperature can be controlled through matching. As for the first confirmation, we explained it from the following three aspects: First, we reviewed the literature and found that either P188 or P407 alone would not have an impact on organisms (1, 2). Secondly, we use it on the surface of the wound surface. It is more difficult for large molecules to enter the body and participate in the circulation than small molecules. Finally, we also injected the mice subcutaneously. After dissolution, we sliced the skin of the mice and observed that no other discomfort was found, so we chose the mixed material to treat the wound. This part of the supplement is modified at:(Line73-76)

2.Please confirm that Nr-CWS was successfully loaded inside the hydrogel.

Response：Sorry we didn't explain that clearly. First of all, we conducted comprehensive detection of the blank gel and the gel coated with the drug in Fig1. We found that the drug Nr-CWS had been encapsulated in the pores by electron microscopy, which was also verified by subsequent release experiments. Therefore, we believe that Nr-CWS was successfully incorporated into the gel. The location is marked (Line 246-247)

3. The biocompatibility of the hydrogel needs to be confirmed before in vivo animal experiments.

Response：As mentioned earlier, we have studied biocompatibility from three aspects. Firstly, we read the literature and found that both P407 and P188 alone have been demonstrated in new aspects of biocompatibility, which will not affect normal cell tissues (3, 4). Secondly, we tried to find the potential danger by subcutaneous injection of mice, but we did not find that these macromolecules would affect normal tissues by observing the HE staining of skin wounds (FigS2). Thirdly, we believe that when gel with drugs is applied to the wound surface, macromolecular gel is more difficult to enter the body than small molecular substances, so the impact will be smaller. And after 12h, we will clean the residual gel. From the above point of view, we believe that its biocompatibility has been well certified. This revision is placed on Line: (Line73-76)

4.Diabetic wound healing is a complex and continuous process. The manuscript indicates that Nr-CWS in the hydrogel releases about 60% of the original within 12 h. This does not seem to meet the need for sustained wound repair. Once the release of Nr-CWS in the hydrogel is terminated, how the hydrogel promotes wound repair?

Response：We apologize for the misunderstanding caused by our rough explanation. As is known to all, in the treatment of diabetic wounds, the wound microenvironment is very poor. In the long treatment cycle, we must carry out debridement regularly to ensure a clean and tidy wound. Therefore, we found that a clinical cycle of 12h is the most effective treatment plan, and a 12h wound cleaning has a good effect on both the patient and the wound healing. Therefore, this gel is also released once for 12h, during which the gel will not only release the drug but also provide good protection to the wound. However, the delivery environment of diabetic wound and gel in the body is different. Diabetic wound will not only have a lot of exudates, but also contact with outside air will cause harmful substances to affect wound healing. Therefore, this gel can absorb exudates and physically block harmful substances from the outside to accelerate wound healing and wound repair.We put the explanation in (Line365-370) of the article.

5.Whether collagen deposition at the wound site after hydrogel treatment causes excessive fibrosis of the wound, explain them.

Response:The question you raised is also the problem we met in our original study. Whether collagen fibers are hyperplasia is mainly to observe whether the arrangement of collagen fibers is disordered. When the staining is dark and the arrangement of collagen fibers is chaotic, we will consider this part as collagen fiber hyperplasia. Through the experiment, we can find that the growth and arrangement of collagen fibers are very regular after the use of gel, so we can conclude that there is no problem of excessive fiber hyperplasia.

6.Why did the author not continue the drug release test for Nr-CWS release performance?

Response:Thank you very much for your question, as pointed out above. Due to the particularity of the diabetic environment, wound cleaning is required regularly to ensure the best effect. However, our drug use time is just once 12h administration frequency, so we did not continue to evaluate the release performance of the drug.

7.In the manuscript, the experimental results are described excessively, and the discussion of scientific problems is lacking. The author had better analyze the scientific problems in the manuscript in detail. In addition, the novelty of this work still needs to be highlighted in the introduction and conclusion.

Response:Our team is very sorry for the confusion in the reading. Here, our team has carefully deleted and supplemented these parts, hoping to get your approval.Lin97-103 and Line384-388 and Line 416-426.

Reviewer #5: In manuscript ‘Smart Thermosensitive Poloxamer Hydrogels Loaded with Nr-CWs for the Treatment of Diabetic Wounds’, the authors designed a P188/P407 poloxamers hydrogel mixed with Nr-CWS to promote angiogenesis and wound repair. This paper will be of interest in readers working in the field of Full-thickness skin defect repair. Some revisions are required to make it appropriate for publication, and the issues related to these revisions are listing below:

Response:Thank you very much for your recognition of our team's work. The whole team is very encouraged to receive your comments. We believe that there is still a long way to go in the treatment of diabetes wounds, but we hope that our work can give some guidance to future generations. Hope to solve the problem of diabetic wound healing as soon as possible.

Q1. There are not enough data to explain the characteristics and innovations of this study.

Response:We are very sorry that we did not emphasize this point in the course of the article writing and the experiment. We will write the introduction and discussion section again. I have focused on replying to your question. We hope that your approval of our supplementary deletion can be obtained, and this article will be published in plos one.Lin97-103 and Line384-388 and Line 416-426.

Q2 In ‘The slow release prolongs the duration of the drug's effect on the wound in vitro and in vivo’ why did only test the drug’s release in 12 hours? More time points should be tested.

Response:We have this problem in our research. The most true state is that complex diabetic wounds are associated with exudation and external physical stimulation on a daily basis, resulting in slow healing problems. However, no matter which preparation of material is applied on the limited wound surface, it is impossible to completely solve these problems. So dressing change debridement has become a problem to be solved at this stage. In the long process of wound healing. Wounds cannot be solved with a single application. So we noticed this in the clinical transformation process and came up with this gel.

Q3. We found no description of the degradation performance of poloxamers hydrogels.

Response:The degradation performance of this gel is beyond doubt, on the one hand, single P188 and P407 have been proven (1,2). Secondly, it is difficult for macromolecular gels to enter the body from skin wounds, and we have proved in cell experiments that they do not have toxic effects on cells. Therefore, we believe that this gel is safe and reliable. In the use cycle once every 12h. The main function of our macromolecular gel is still wound protection and drug release.we did not find that these macromolecules would affect normal tissues by observing the HE staining of skin wounds (FigS2). So it's almost never metabolized in the body and it doesn't affect safety.I'm sorry, we have explained this problem. Put the modified part in Line (Line73-76)

Q4. In ‘Fig.4D’, we didn’t see the scale bar in both 6 and 14 days

Response:I am very sorry that we ignored this problem in our previous work. We have revised it and uploaded it again.

Q5. In ‘Fig.4F’, we can hardly see the difference in collagen deposition between four groups in 14 days.

Response:We can tell by the color and arrangement of the collagen fibers. We found that the structure arrangement of Nr-CWS gel was more orderly than that of blank gel. That means it heals better. However, the color of diabetic wounds was significantly lighter than that of blank gel, indicating that the density and growth of collagen fibers were far lower than that of gel group. We believe that the repair of collagen fibers is quite obvious in this comparison.

Q6. There were still format problems. For example, the title ‘Western blot analysis’ should be bold. ‘Establishment of Diabetic diabetic wound model’ should be deleted ‘Diabetic’; the ‘°C’ in line 106 and 159 were different and we didn’t see a space between ‘0.5h,shaken’ in line 160 so on.

Response:I'm really sorry that we have rearranged and modified all the parts.We make all °C in the text the same See in Line 116, 117, 147 and 171 and so on.

Reference

1. Tan MSA, Pandey P, Lohman RJ, Falconer JR, Siskind DJ, Parekh HS. Fabrication and Characterization of Clozapine Nanoemulsion Sol-Gel for Intranasal Administration. Mol Pharm. 2022;19(11):4055-66.

2. Kjar A, Wadsworth I, Vargis E, Britt DW. Poloxamer 188 - quercetin formulations amplify in vitro ganciclovir antiviral activity against cytomegalovirus. Antiviral Res. 2022;204:105362.

3. Moiseev RV, Steele F, Khutoryanskiy VV. Polyaphron Formulations Stabilised with Different Water-Soluble Polymers for Ocular Drug Delivery. Pharmaceutics. 2022;14(5).

4. Kamel R, El-Wakil NA, Elkasabgy NA. Calcium-Enriched Nanofibrillated Cellulose/Poloxamer in-situ Forming Hydrogel Scaffolds as a Controlled Delivery System of Raloxifene HCl for Bone Engineering. Int J Nanomedicine. 2021;16:6807-24.

---

## [Editor Report · Decision Letter 2]

13 Dec 2022

Smart Thermosensitive Poloxamer Hydrogels Loaded with Nr-CWs for the Treatment of Diabetic Wounds

PONE-D-22-22609R2

Dear Dr. Jin,

We’re pleased to inform you that your manuscript has been judged scientifically suitable for publication and will be formally accepted for publication once it meets all outstanding technical requirements.

Kind regards,

Bing Xu, PhD

Academic Editor

PLOS ONE
---

## [Editor Report · Acceptance letter]

21 Dec 2022

PONE-D-22-22609R2 

Smart Thermosensitive Poloxamer Hydrogels Loaded with Nr-CWs for the Treatment of Diabetic Wounds 

Dear Dr. Jin:

I'm pleased to inform you that your manuscript has been deemed suitable for publication in PLOS ONE. Congratulations! Your manuscript is now with our production department. 

Kind regards, 

on behalf of

Dr. Bing Xu 

Academic Editor

PLOS ONE